# Use of the Single Particle Soot Photometer (SP2) as a pre-filter for ice nucleation measurements: Effect of particle mixing state and determination of SP2 conditions to fully vaporize refractory black carbon

Gregory P. Schill[1], Paul J. DeMott[1], Ezra J.T. Levin[1], and Sonia M. Kreidenweis[1]

[1]Department of Atmospheric Science, Colorado State University, Fort Collins, 80521, United States of America

*Correspondence to*: Gregory P. Schill (gpschill@atmos.colostate.edu)

**Abstract.** Ice nucleation is a fundamental atmospheric process that impacts precipitation, cloud lifetimes, and climate. Challenges remain to identify and quantify the compositions and sources of ice nucleating particles (INPs). Assessment of the role of black carbon (BC) as an INP is particularly important due to its anthropogenic sources and abundance at upper tropospheric cloud levels. The role of BC as an INP, however, is unclear. This is, in part, driven by a lack of techniques that directly determine the contribution of refractory BC (rBC) to INP concentrations. One previously developed technique to measure this contribution uses the Single Particle Soot Photometer (SP2) as a pre-filter to an online ice nucleating particle counter. In this technique, rBC particles are selectively heated to their vaporization temperature in the SP2 cavity by a 1064-nm laser. From previous work, however, it is unclear under what SP2 conditions, if any, that the original rBC particles were fully vaporized. Furthermore, previous work also left questions about the effect of the SP2 laser on the ice nucleating properties of several INP proxies and their mixtures with rBC.

To answer these questions, we sampled the exhaust of an SP2 with a Scanning Mobility Particle Sizer and a Continuous Flow Diffusion Chamber. Using Aquadag® as an rBC proxy, the effect of several SP2 instrument parameters on the size distribution and physical properties of particles in rBC SP2 exhaust were explored. We found that a high SP2 laser power [930 nW/(220-nm PSL)] is required to fully vaporize a ~0.76 fg rBC particle. We also found that the exhaust particle size distribution is minimally affected by the SP2 sheath-to-sample ratio; the size of the original rBC particle, however, greatly influences the size distribution of the SP2 exhaust. The effect of the SP2 laser on the ice nucleation efficiency of Snomax®, NX-illite, and Suwannee River Fulvic Acid was studied; these particles acted as proxies for biological, illite-rich mineral dust, and brown carbon INPs, respectively. The original size distribution and ice nucleation efficiency of all non-rBC proxies were unaffected by the SP2 laser. Furthermore, the ice nucleation efficiencies of all proxies were not affected when externally mixed with rBC. These proxies, however, always show a reduction in ice nucleating ability when internally mixed with rBC. We end this work with recommendations for users who wish to use the SP2 as a prefilter to remove large rBC particles from an aerosol stream.

## 1 Introduction

Black carbon (BC) aerosol, emitted through incomplete combustion of carbon-based fuels, affect the Earth's climate directly by scattering and absorbing incoming solar radiation and outgoing terrestrial radiation. BC is a short-lived climate forcer and its direct effect is estimated to exert a positive net radiative forcing; thus, it is a prime candidate for near-term climate mitigation and has been implicated as one of the so-called mitigation "wedges" (Grieshop et al., 2009). In addition to its direct radiative effects, BC aerosol can also affect the Earth's climate indirectly by serving as nuclei for cloud-droplets or ice particles in clouds. To fully assess the potential of BC regulation as an effective near-term climate wedge, however, its radiative forcing due to both

its direct and indirect effects must be considered. Currently, the indirect effect due to BC, especially its role as an ice-nucleating particle (INP), is highly uncertain. For example, the sign of the BC ice-cloud forcing is, at present, not known to be positive or negative (Bond et al., 2013). One contributing factor to this uncertainty is that the direct contribution of BC to INP concentrations is not well constrained.

Previous studies that determined the contribution of BC to INP concentrations exhibit conflicting results. For example, mountain-top studies at the high-Alpine research station Jungfraujoch found that BC particles were enhanced in the residuals of ice crystals from mixed-phase clouds (Cozic et al., 2008); subsequent studies at the same site, however, determined that BC were very poor INPs (Kamphus et al., 2010; Kupiszewski et al., 2016). In addition, several aircraft studies have shown a correlation between ice cloud glaciation and BC aerosol concentration (Pratt et al., 2009; Stith et al., 2011; Twohy et al., 2010); however, the

specific material responsible for ice nucleation was not definitively determined. Ground-based studies also indicate that biomass burning may be a variable source of BC INPs (McCluskey et al., 2014; Prenni et al., 2012), and laboratory burns have shown that BC can, for some fuel types, contribute to large fractions of INPs (Levin et al., 2016). In other laboratory studies, however, BC from an off-road diesel engine did not contribute appreciably to INP concentrations in diesel exhaust, even when aged to 1.5 equivalent photochemical days (Schill et al., 2016). Consequently, it is becoming evident that both fuel type and combustion

conditions can greatly affect the ice nucleation efficacy of BC. Given these dependencies, the direct contribution of rBC to ambient INP concentrations remains highly speculative.

          A main factor contributing to this speculation, especially in field studies, is that few analytical techniques can determine the contribution of BC to ambient INP concentrations. Currently, the only technique to do so directly and in real-time uses a Single Particle Soot Photometer (SP2) as a pre-filter to an online ice nucleation counter, the Continuous Flow Diffusion Chamber [CFDC,

(Levin et al., 2014)]. In the SP2 pre-filter, refractory black carbon (rBC) is vaporized by a high-intensity 1064-nm laser; aerosol that do not absorb at 1064 nm, which encompasses most ambient non-rBC aerosol, pass through the cavity unaltered. Thus, by toggling the laser on and off while sampling the SP2-exhaust with an online ice nucleation counter, the direct contribution of rBC to INP concentrations can be determined.

          The initial study of this technique (Levin et al. 2014) exposed two processes that could confound SP2-CFDC results. First,

rBC incandescence and subsequent vaporization formed new, sub-100 nm particles. These new particles were postulated to form in the SP2 exhaust via homogeneous nucleation of rBC vapors; however, it was unclear if the original rBC particle was fragmented, partially ablated by the laser, or if full vaporization occurred. Fragments and partially ablated particles could confound SP2-CFDC results since both could still be active as INPs. In the second confounding factor, incandescing rBC destroyed ice-nucleation-active sites on kaolinite when rBC was physically attached to, or internally mixed with, kaolinite. This confounds SP2-CFDC results by

overestimating the contribution of rBC to INP concentrations. How important this factor is for other mixtures of rBC and INP proxies, as well as its relative importance for both internal and external mixing scenarios remains unknown.

          The purpose of this work was two-fold. Our primary interest was determining if mixtures of rBC and proxies of biological particles, illite-rich dust, or brown carbon (BrC) are affected by the SP2 laser. To do so, we measured the size distribution and ice nucleation activity of SP2 exhaust from sampling Snomax®, NX-illite, and Suwanee River Fulvic Acid (SRFA) particles,

respectively. These proxies were then internally and externally mixed with rBC and the mixture's ice nucleation efficiencies were determined. To correctly interpret the mixing-state experiments, we had a secondary interest to define under what SP2 conditions are rBC particles fully vaporized.

## 2 Experimental

The complete experimental setup is shown in Figure 1. Particles were aerosolized from solution using one or two medical nebulizers and introduced into a small, 10 L mixing chamber. The aerosol in the mixing chamber could be diluted with dry, particle-free air to control total aerosol concentrations. Particles sampled from the mixing chamber were dried to RH <5% using two annular diffusion dryers, and size selected using a Differential Mobility Analyzer (DMA). At the outlet of the DMA, the flow was split to a Condensation Particle Counter (CPC) and the SP2. Since the SP2 sheath flow dilutes aerosol concentrations in the SP2 exhaust, the CPC measured the undiluted particle concentration. Dilution corrections were always applied to the SP2 exhaust measurements. SP2 exhaust was measured either by the CFDC or a Scanning Mobility Particle Sizer (SMPS, TSI 3936L10). When operating the SMPS at 1 Lpm, a makeup flow of 0.5 Lpm particle-free air was added to keep the flows as close as possible to the CFDC flow of 1.5 Lpm.

### 2.1 Single Particle Soot Photometer

The SP2 consists of an intense, intracavity 1064-nm laser and four optical detectors used to determine the broadband incandescence, narrowband incandescence, scattering, and time-dependent location (split detector). The laser was tuned to the TEM 0,0 mode, resulting in a Gaussian-profiled beam. An aerosol flow is introduced orthogonal to the path of the beam and plane of the laser and detectors. Particles that do not have appreciable absorption at 1064 nm will only scatter light and be detected by the scattering and time-dependent detector. Those particles that do appreciably absorb 1064-nm light, which in ambient aerosol particles is primarily rBC (Stephens et al., 2003), will heat up and emit visible thermal radiation, or incandesce. When operating at an absolute laser power (see section 2.2) of at least 450 nW/(220-nm PSL), the SP2 has an rBC lower detection limit of ~0.7 fg (Schwarz et al., 2010). The upper rBC detection limit depends on the sensitivity of the incandescence detectors; for the SP2 used here (Droplet Measurement Technologies, 8-channel version, #052) it was ~165 fg. Detection limits for the scattering detector will depend on laser power, optical alignment of the system, and the sensitivity of the detectors. Thus, this number will vary between instruments and between optical cleanings and alignments of the same instrument.

The SP2 scattering and rBC incandescence responses were calibrated with ammonium sulfate and Aquadag® (Acheson, lot #9054), respectively. Aquadag is an aqueous-based colloidal dispersion of ultrafine graphitic carbon. The total-carbon content of Aquadag® is 90-93%, while the organic-carbon content is 7-10% (Gysel et al., 2011). Although it has been shown that denuding Aqaudag particles at 400 °C increases their SP2 response by 15% (Laborde et al., 2012), we did not use a thermal denuder in our calibrations, primarily because it is customary in the SP2 community not to do so. The SP2 data and calibration were processed using the Paul Scherrer Institute SP2 Toolkit version 4.100a. Both Aquadag® and ammonium sulfate were size selected using a DMA. The mobility-diameter-selected Aquadag® was converted to a physical diameter using the effective density curves found in Gysel et al. (2011). The scattering response was fitted to the mobility diameter of ammonium sulfate using Mie scattering calculations; the refractive index of $(NH_4)_2SO_4$ was assumed to be $1.51m + 0i$.

### 2.2 Calculating SP2 Absolute Laser Power

To quantitatively assess the laser power needed to completely vaporize rBC particles in the SP2, we calculated the absolute laser power using calibration PSLs. The SP2 inherently measures laser power qualitatively using a leakage detector, which is useful to monitor the consistency of the laser alignment. It is difficult, however, to relate this measurement to absolute laser power because the finesse of the Nd:YAG crystal face as well as the intra-cavity mirror are unknown. Furthermore, it is difficult to draw correlation between the leakage detector voltage and the absolute laser power (Figure S1). To account for this, as well as differences in the avalanche photo diode (APD) gain between SP2 instruments and versions, Schwarz et al. (2010) developed a method to calculate

the absolute SP2 laser power independent of these variances. There, reference PSLs were used to determine the peak optical power to the scatter detector APD. The peak power incident on the APD from a 220-nm PSL particle, P, can be calculated from the peak voltage produced by the scatter detector:

$$P\left(\frac{nW}{220-nm\ PSL}\right) = \frac{S \cdot 2.44_{(mV)}}{[R1_{\Omega}]\ [G2]\ [A_{(A/W)}]} * 10^6,$$

(1)

where S is the modal scattering-peak height from a 220-nm PSL sampled in units of 2.44 mV steps (the resolution of the SP2's high speed analog to digital converter); R1 is the impedance of the resistor used in the current-to-voltage conversion for the first stage of gain in the SP2 APD board (49.9 kΩ for SP2 #052); G2 is the gain of amplification (2 and 20.1 for the low and high gain on SP2 #052, respectively); and A is the sensitivity of the APD (12 A/W for SP2 #052). It is important to note that the APD response depends on both its bias voltage and temperature. Thus, the SP2 should be used within ±5 °C of the temperature used for the absolute power calculation. For most experiments, 220-nm PSL were used, but in some cases 300-nm PSL were used. The peak power scattered by PSLs of other sizes can be scaled to 220 nm using Mie theory as outlined by Schwartz et al. (2010).

### 2.3 SP2 Attached-Type Particle Approximation

To determine the number of attached-type particles, we monitored the half-decay positions of both the incandescence and scattering signals. When an rBC particle coated with non-refractory material traverses the SP2 laser, it both scatters light and starts to incandesce (Figure S2). As the incandescent core is heated to its vaporization temperature, the particle shrinks since the core is both boiling off non-refractory material and vaporizing. Thus, the scattering signal eventually falls off precipitously. It is important to note that the incandescent detector has a size range of ~90-500 nm, but the scattering detector has a much more limited range of ~200-400 nm. Due to these size-range differences, the half-decay position of the scattering peak always appears prior to the half-decay position of the incandescent peak for a coated particle. When an rBC particle is attached to a refractory particle such as dust, the attached rBC particle is vaporized, but the dust particle passes through the laser unaffected or only partially ablated (Figure S3). Thus, in this scenario, the half-decay position of the scattering peak always appears after the half-decay position of the incandescent peak. To approximate the number of attached-type particles, we take the difference between the scattering peak's half-decay position and incandescent peak's half-decay positions. Those that have positive differences we defined as "attached-type" particles. The range of attached rBC masses is limited to 0.7 to 25.4 fg (90-300 VED diameter, 1.8 g cm$^{-3}$). This corresponds to 120-450-nm Aquadag® particles. For pure Aquadag® particles, the fraction of attached type particles was 0.03. This approach is less rigorous than the time-dependent cross section calculation outlined by Moteki et al. (2014), but is also less computationally expensive. Thus, this technique can analyze a wide range of attached rBC masses simultaneously.

### 2.4 Continuous Flow Diffusion Chamber

The CFDC has been described previously in detail (DeMott et al., 2015; Rogers et al., 2001). Briefly, the CFDC consists of two, vertically mounted concentric copper columns that have been chemically treated to be wettable by liquid water. Each column is temperature-controlled using refrigeration compressors. The annular gap between the two bare columns is 1.1 cm. When taking a cross-section of the columns, the inner and outer column form the inner and outer walls, respectively. To form a thin layer of ice on each wall, the annular volume is flooded with water while both wall temperatures are held at -27 °C. After icing, an aerosol lamina is tightly focused between the walls by directing the sample flow to pass through a cylindrical knife edge, surrounded by dry and particle-free sheath flows representing 85% of the total flow. By controlling the temperature of each ice-coated wall, the temperature and RH of the aerosol lamina can be precisely controlled.

For this study, the temperature and RH inside the aerosol lamina were set to approximately -30 °C and 105%, respectively. Under these water-supersaturated conditions, most aerosol particles will activate into droplets. Those droplets that contain INPs active at -30 °C or warmer will nucleate ice. The bottom third of the CFDC column is a droplet evaporation section, where conditions at the aerosol lamina are adjusted to ice saturation and, therefore, at water sub-saturated conditions. Thus, particles that activated into droplets, but have not nucleated ice, evaporate and return toward their original size. All particles >500 nm are detected by an Optical Particle Counter (OPC), and ice particles are assumed to be larger than a size-calibrated threshold (3.0 µm). To ensure that large aerosol particles were not miscounted as ice crystals, a two-stage 2.4 µm impactor was placed at the CFDC inlet.

### 2.5 Scanning Mobility Particle Sizer

To determine the size distribution of the SP2 exhaust, a TSI Scanning Mobility Particle Sizer (SMPS, Model 3080 DMA, Model 3010 CPC) was used. To access a wide size range, a low sheath-to-sample ratio (5:1) was employed. By passing the particles through a Po-210 neutralizer, aerosol charge equilibrium was reached immediately prior to the DMA inlet. Scan times were 180 seconds, and particles were corrected for multiple charging and diffusion in TSI Aerosol Instrument Manager.

### 2.6 Transmission Electron Microscopy

Electron micrographs of the SP2 exhaust when sampling Aquadag® were taken with a JEOL JEM-2100F Transmission Electron Microscope. All imaging was done at 200 kV. Images were captured with a Gatan Ultrascan 1000XP CCD camera using 2048 X 2048 pixels. SP2 exhaust particles were collected on a carbon film supported on 200 mesh copper grids. Grids were placed in the center of a 47 mm Nuclepore® filter and placed in a Pall stainless steel inline filter holder.

### 2.7 Mixed-Particle Generation

To generate internally or externally mixed rBC and INP proxy particles, we wet-generated rBC and INPs from one or two medical nebulizers, respectively. The INP types were chosen to represent a proxy for biological particles (Snomax®), an illite-rich mineral dust (NX-illite), and BrC (SRFA). In the "internal mixing" scenario, Aquadag® and an INP type were combined in the same nebulizer. Each constituent was added to the nebulizer with ultra-high purity water at 0.05 wt%. Particles were size selected at 500 nm with a DMA, but it is expected that the amount of rBC varies from particle to particle. In a second scenario, which we call "external mixing," 0.1 wt % Aquadag® and an INP type are nebulized by separate nebulizers. The separate nebulizer aerosol streams were then added separately to the 10 L mixing chamber. While coagulation inside the mixing tank is possible, it is assumed that the concentration of coagulated particles is small. Typical concentrations during the external mixing scenario were ~100-200 $cm^{-3}$ of 500-nm size-selected particles.

### 2.8 Sampling SP2 Exhaust

To sample the SP2 exhaust with another aerosol instrument, we modified the exhaust-line of the SP2. The general method has been previously described by Levin et al. (2014) and Aiken et al (2016), and has been expanded in this work. Under normal operation, the SP2 operator sets only the sample and sheath flow. Two separate pumps control the exhaust and the sheath/purge flows; the sheath and purge flows confine the aerosol stream to the center of the laser and protect the laser optics, respectively. The sheath flow is regulated by a mass flow controller (MFC); the purge flow, also regulated by an MFC, is slaved to the sheath flow set point and is set to be approximately 1/3 of the sheath flow. The exhaust flow is regulated by a proportional valve, which

is controlled by a proportional–integral–derivative (PID) loop. The PID controller increases/reduces the exhaust flow to match the sample flow to its set point.

In the modified setup, we isolate the SP2 exhaust pump and use an additional aerosol instrument to pull the SP2 exhaust flow. Thus, the SP2 sample flow is no longer regulated by the PID loop, and therefore the sample flow simply equals the modified exhaust flow minus the sheath and purge flow. When switching between the SP2 exhaust pump and the additional aerosol instrument, special care must be taken to ensure that the exhaust is always being pulled by a flow, otherwise the operator runs the risk of contaminating the intra-cavity optics. Contamination often results in a catastrophic loss in absolute laser power that can only be remedied by cleaning and aligning the entire optical system. To avoid this, a double three-way-valve system was added to the exhaust port of the SP2 cavity (Figure 1). Here, the SP2 exhaust flow can be (1) completely isolated, (2) pulled by only the SP2 exhaust pump, (3) pulled by an external aerosol instrument, or (4) pulled by both the SP2 exhaust pump and external instrument, all by toggling the 3-way valves. To switch the SP2 exhaust without contaminating the optics, the valves are operated such that the additional aerosol instrument momentarily pulls flow in addition to the SP2 exhaust pump (situation 4). Only then is the SP2 exhaust pump isolated from the exhaust line.

Under the modified setup, the SP2 sample flow is not restricted or controlled; however, the sample flow is measured using a laminar flow element. Thus, the purge, sheath, and sample flows are constantly monitored. This allows us to determine the dilution ratio of the sample flow caused by the sheath and purge flows. Constant monitoring of the dilution ratio is especially important when taking ambient measurements and when comparing to aerosol measurements not sampling from the SP2 exhaust.

## 3 Results

The primary question of this work was to determine if particle mixing state could confound results when using the SP2 as a pre-filter for ice nucleation measurements. To answer this, we first had to determine (1) under what SP2 instrumental conditions are rBC particles fully vaporized, (2) what are the physical properties of the exhaust, and (3) if the SP2 laser affected pure INP proxies of biological particles, illite-rich dust, and BrC. Only then could we address the primary goal of determining the effect of the SP2 laser on mixed rBC-INP particles.

### 3.1 Dependence of Aquadag® SP2 Exhaust Physical Properties on SP2 Operational Parameters

To determine the effect of several SP2 operational parameters on the physical properties of rBC SP2 exhaust, we measured SP2 exhaust with the SMPS while sampling the incandescent material Aquadag®. To ensure the applicability of these results to other SP2 instruments, we first determined the absolute laser power needed to fully vaporize rBC particles in the SP2 laser. As noted by Schwartz et al. (2010), variations exist between each SP2 and even within one SP2 between separate optical alignments; thus, the absolute laser power in units of nW/(220-nm PSL), and not the voltage from the leakage detector, must be known. As shown in Figure 2, 125 nm (0.76 fg) size-selected Aquadag® particles pass through the SP2 cavity unaffected when the laser is off. At 130 nW/(220-nm PSL), the shoulder of a new particle peak arises at ~25 nm. The new particles are formed from either homogeneous nucleation or fragmentation of the original particle. At 370 nW/(220-nm PSL), the shoulder of the new particle peak is ~35 nm and the original particle is slightly ablated. As the laser power is increased, the resulting particles in the SP2 are further ablated, but the shoulder of the new particle peak plateaus at ~35 nm. At 930 nW/(220-nm PSL), the original particle is either fully vaporized, or ablated to a size below the new particle shoulder. Thus, we assume that the SP2 laser power is sufficient to completely remove the initial rBC particle at 930 nW/(220-nm PSL). Electron microscopy results that support this are shown below. Thus, for future studies using the SP2 as a pre-filter for rBC, we recommend that the SP2 laser be, at minimum, 930 nW/(220-nm PSL). It

is interesting to note that this minimum power to vaporization is over twice the value outlined by Schwartz et al. (2010) needed to detect 0.7 fg rBC particles with uniform efficiency. Thus, particles can be detected in the SP2 with uniform detection efficiency without vaporizing them, similar to pulsed laser-induced incandescence techniques (Michelsen et al., 2007, 2015). For all subsequent SP2-exhaust results, data was taken at an absolute laser power of at least at 930 nW/(220-nm PSL).

INP concentrations are exceedingly low [roughly 1 in $10^5$ particles nucleates ice in the free troposphere at an average temperature of approximately -20 ºC, (Rogers et al., 2001)]. In the SP2 cavity, the sheath and purge flows dilute the original aerosol concentration in the exhaust, further decreasing INP concentrations. Thus, when using the SP2 as a pre-filter for online INP measurements, we attempt to keep the SP2 sample flow high, and, therefore, the sheath-to-sample ratio low. The purpose of the sheath flow is to confine the aerosol stream to the center ¼ of the Gaussian profile of the laser beam. Thus, it is important to test

whether a low sheath-to-sample ratio affects the ability of the SP2 laser to vaporize rBC particles. In particular, we are interested in knowing if low sheath-to-sample ratios could enable the sample to flow outside of the laser beam. As shown in Figure 3, the sheath-to-sample ratio does not appreciably affect the aerosol size distribution in the SP2 exhaust. It is important to note that in these experiments, the particle size and concentration were kept constant at 125 nm and 600 cm$^{-3}$, respectively. As shown, even for sheath-to-sample ratios as low as 2.5:1, 125 nm Aquadag® particles are completely vaporized (below 50 nm). There is a slight

trend for lower sheath-to-sample ratios to shift the shoulder of the new particle distribution to larger sizes. The time for particles to traverse the SP2 laser is dictated by the total flow, which was kept constant at 1.5 Lpm by the SMPS and dummy flow. With a constant laser-traversing time as well as constant particle size and concentration, lower sheath-to-sample ratios enable a larger flux of rBC through the laser. It is likely that larger fluxes of rBC mass through the laser are causing the shoulder of the new particle peak to shift to slightly larger sizes.

If a larger flux of mass from a size-selected rBC particle caused a shift in the new particle formation peak, we also expect larger rBC particles to shift the new particle peak shoulder to larger sizes. Thus, we ran the SP2-SMPS for size-selected rBC particles from 90 to 500 nm (Figure 4). In these experiments, the absolute concentration of size-selected particles was kept at approximately 1000 cm$^{-3}$ and the sheath:sample was 4.9:1. As shown in Figure 4, the number size distribution of the rBC SP2 exhaust depends on the initial rBC size. As the size of the initial rBC particle decreases, the tail of the new particle size distribution

moves to smaller sizes. Interestingly, even the 100-nm (0.41 fg) particle, which is below the uniform detection limit of the SP2, does not pass through the SP2 cavity unaffected. This indicates that the particle is being heated, at least partially, to its vaporization temperature. Thus, the limitation for the SP2 at these low masses, at least for Aquadag®, may be the SP2 incandescence detector. At large initial rBC sizes, the shoulder of the new particle peak can extend up to 100-nm. Fortunately, insoluble particles smaller than 100 nm are not expected to participate in ice nucleation (Marcolli et al., 2007). Additionally, the mode of ambient rBC mass

distributions are smaller than 500 nm, and therefore the shoulder of the new particle peak is expected to be much lower than 100 nm. For example, an SP2-SMPS measurement of ambient air at Colorado State University when rBC loadings were 3.2 ± 1.4 cm$^{-3}$ (20150330) is shown in Figure S4. The "laser on" size distribution of the ambient SP2 exhaust deviates from the "laser off" size distribution at ~60 nm. Thus, the new particles formed in ambient SP2 exhaust are not expected to participate in ice nucleation relevant to mixed-phase clouds. Concurrently, all previous tests and deployments of the SP2-CFDC have never found increased

INP concentrations.

### 3.2 Physical Properties of rBC SP2 Exhaust

       The combined results from our SMPS tests indicate that, above a laser power threshold (930 nW/220 PSL), the initial rBC is completely vaporized; using the SP2-SMPS, however, only provides indirect evidence of this. To address this directly, we took TEM images of 500-nm, size-selected Aquadag® particles passed through the SP2 with the laser on and off. As shown in

Figure 5a, Aquadag® particles are non-spherical, crumpled graphitic sheets. At higher resolution, the graphitic structure (0.335 nm spacing) of the Aquadag® emerges (Figure 5b). When the SP2 laser is turned on, 500 nm Aquadag® particles are vaporized and new particles form in the SP2 laser cavity. The number concentration of the initial 500 nm particles and the collection times for the laser on and laser off experiment were similar. A comparison of Figures 5a and 5c shows that the new particles are much more abundant, which agrees with the SP2-SMPS results. When looking at the new particles on the same scale as Figure 5b, the graphitic structure of the original Aquadag® particles is not apparent (Figure 5d). An amorphous structure indicates that the new particles are not fragments of the original Aquadag® particles. Thus, this affirms that the particles are formed from homogeneous nucleation of the vapors formed during the vaporization of the original Aquadag® particle.

### 3.3 Effect of SP2 Laser on Snomax®, NX-Illite, and SRFA Size Distributions

Previously, the effect of the SP2 laser on both PSLs and Arizona Test Dust (ATD) were determined; in the ambient atmosphere, however, several additional aerosol types may act as INPs. These aerosol types include biological particles, illite-rich dust, and BrC. Thus, it is important to know if the SP2 laser alters their size or ice nucleation activity. As a proxy for biological INPs, we used Snomax®, a lyophilized preparation of *Pseudomonas Syringae* used commercially for inducing snow at ski areas. Although Snomax® is not expected to absorb 1064-nm light, biological particles are heat labile and could be irreversible damaged in the SP2 cavity if absorbing material was present. As shown in Figure 6a, the SP2 laser does not affect the size distribution of Snomax®. SRFA (International Humic Substances Society; 1S101F) is dissolved organic carbon isolated from Suwannee River aquatic samples by filtration, concentration on an XAD-8 resin column, separation from humic acids at pH < 2, de-salting, neturalization, and freeze-drying (Averett et al., 1994). In this work, it is used as a proxy BrC, which is operationally defined as an organic carbon species that has a linear increase in absorption from the visible to the near-UV (Laskin et al., 2015). Because of the possibility of a non-zero absorption in the IR region, it is important to determine whether BrC is affected by the SP2 laser. As shown in Figure 6b, the size distribution of SRFA is unaffected by the SP2 laser, indicating that this BrC proxy, which is highly colored, does not absorb appreciably at 1064 nm.

Unlike the Snomax® and SRFA, the mineral dust sample (ATD) used by Levin et al. (2014) had a small incandesce signal in the SP2, which resulted in a new-particle peak similar to rBC exhaust. Unlike rBC, however, its original size distribution was only minimally affected. The mineralogical composition of transported atmospheric dust samples has recently been shown to be more like illite-rich dust than ATD (Broadley et al., 2012). Thus, it is important to test if illite-rich dust samples have similar incandescent behavior in the SP2, and if the laser affects the original size distribution. For the ice nucleating "internally mixed" experiments described below, the NX-illite particles needed to be wet generated. In the past, several publications have shown that wet-generation of dust particles affects their cloud condensation nuclei activity and size distributions. For example, Koehler at al. (2009) found that the hygroscopicity parameter ($\kappa$) of ATD was 0.02 when generated dry from a fluidized bed and 0.35 when generated wet from an atomizer. Garimella et al. (2014) found that dry-generated ATD, illite, and Na-monmorillonite were all primarily >100 nm in mobility diameter, while the exact same dust wet-generated had a significant number fraction of particles <100 nm. It was concluded that these <100 nm particles were likely soluble material that was leached from the dry dust. The size distributions of both dry and wet generated NX-illite SP2 exhaust are shown in Figures 6c and 6d, respectively. As shown, we see that dry and wet-generated particles see similar behavior to Garimella et al. (2014). In both cases, the NX-illite particles were also mildly incandescent. From the SP2 measurements, those particles with measurable incandescence were 0.002% of all by particles by number. Thus, despite the formation of this new particle shoulder in the size distribution spectrum, the original size distribution was insignificantly affected.

**3.4 Effect of SP2 Laser on Mixtures of INP Proxies with rBC**

Since the original size distributions of the INP proxies explored in this work were minimally affected for >99.998% of the particles, we expect that their ice nucleation behavior would also minimally be affected. To ensure this, we first determined the effect of the SP2 laser on the ice nucleation ability of pure proxies by using the CFDC. The CFDC was operated at -30 °C and 105% RH for all measurements. To aid in our analysis, we define the ice nucleation efficiency parameter $\xi_T$ (Petters et al., 2009) as

$$\xi_T = log_{10} \frac{N_{INPs}}{N_{total}}, \qquad\qquad (2)$$

where T is the temperature in Celsius, $N_{INPs}$ are the total number of INPs as determined by the CFDC and corrected for SP2 dilution, and $N_{total}$ is the total number of aerosol as determined by a CPC. The ice nucleation efficiencies of 500-nm size-selected Snomax®, NX-illite, and SRFA are shown in Figure 7. Size-selected Snomax® and NX-illite have $\xi_{-30} > -3$, and all values are in agreement with previous literature (Hiranuma et al., 2015; Wex et al., 2015). The ice nucleation efficiency of SRFA has been studied in the past (O'Sullivan et al., 2014; Wang and Knopf, 2011), but no previous measurements at -30 °C have been taken with an online ice nucleation counter. Thus, these measurements are the first $\xi_T$ values reported for SRFA. As shown in Figure 7, 500 nm SRFA is active as an INP at -30 °C, but it is less efficient than Snomax® and NX-illite by an order of magnitude or greater. Currently, the entity responsible for ice nucleating in SRFA is unknown. SRFA consists of 0.5-1.5 kDa organic macroligands, which are amphiphilic and contain carbonyl, phenol, and ketone moieties. While these masses are well below the expected sizes of most known ice nucleation entities (INE), both dynamic light scattering and small angle neutron scattering indicate that aggregates ~100-500 nm in diameter can form even in dilute solutions of SRFA (Diallo et al., 2005; Palmer and Von Wandruszka, 2001). These aggregates contain macrostructures that could induce ice nucleation. Finally, INP concentrations from 500 nm Aquadag® particles (179.0 cm$^{-3}$) was found to be below the CFDC significance level ($\xi_{-30} < 4.2$; accounting for SP2 dilution). This value is below the previously determined for 600-nm size selected Aquadag® particles (Levin et al., 2014)]. It is unknown what caused this discrepancy; however, it should be noted that Aquadag® has an ash content up to 0.15%. If the ash content is heterogeneously distributed in the stock sample, then Aquadag® solutions from the same stock may contain varying amounts of mineralogical impurities that can act as INPs. Also shown in Figure 7 are the $\xi_{-30}$ pure INP proxies when passed through the SP2 with the laser on. As shown, the $\xi_{-30}$ of the INP proxies are all approximately the same when the laser is both on and off. These results indicate that the laser is minimally affecting the ice nucleation properties of the explored INP proxies, similar to the size distribution results in Section 3.3.

Previously, electron microscopy studies have shown that rBC emitted from wildfires may be internally or externally mixed with both refractory and non-refractory material (China et al., 2013; Pósfai et al., 2003). Furthermore, it has been shown that the SP2 laser can reduce the ice nucleation efficiency of ATD if it is internally mixed with rBC (Levin at el. 2014). Thus, rBC internally mixed with other INPs could artificially enhance our measured rBC contribution to ambient INP concentrations. Additionally, since rBC is heated to its vaporization temperature of 4000 K in the SP2 cavity, it is possible that heat sensitive INPs such as biological particles or BrC may be altered even if they are not physically attached to an rBC particle. Finally, condensation of evaporated organics or vaporized rBC may coat and deactivate externally mixed INPs; while this is a lesser concern for non-biologically sourced liquid organics (Prenni et al., 2009; Schill et al., 2016), a coating of solid, amorphous carbon could cause a significant reduction in ice nucleation ability. To constrain these mixing effects, we ran both external and internal mixtures of rBC with Snomax®, NX-illite, and SRFA and determined the rBC contribution. The rBC contribution is defined as:

$$\text{rBC Contribution} = 1 - \frac{(N_{INP, SP2\ Laser\ On})/(N_{Total})}{(N_{INP, SP2\ Laser\ Off})/(N_{Total})} \qquad (3)$$

where $N_{INP,\ SP2\ Laser\ On}$ is the number or INP measured by the CFDC and corrected for dilution when the sampling the SP2 exhaust with the laser on, $N_{INP,\ SP2\ Laser\ Off}$ is the number or INP measured by the CFDC and corrected for dilution when the sampling the SP2 exhaust with the laser off, and $N_{Total}$ is the total number concentration of aerosol measured by the CPC upstream of the SP2, As shown in Figure 8, the rBC contribution to INPs in the external mixtures was < 5% for Snomax® and NX-illite and < 20% for SRFA. Interestingly, non-rBC particles do not seem to serve as a condensational sink for the vaporized rBC, as their ice nucleation and therefore surface properties are minimally affected. This could be the case for vaporized organic coatings on ambient rBC; however, non-biological, liquid-organic coatings generally do not affect immersion-freezing (Levin et al., 2014). The cause of larger deactivations in rBC-SRFA external mixtures is unknown; however, it was unlikely localized heating, as fulvic acids are expected to be even less heat-labile than proteins such as Snomax®. It may be possible that liquid SRFA particles have a higher propensity to coagulate in the mixing tank with rBC particles than solid Snomax or NX-illite particles; however, we have no evidence to corroborate this statement. Nonetheless, we find that incandescing particles generally do not affect the efficacy of mineral dust and biological INP proxies when they are not physically attached.

When the INP proxies are internally mixed with rBC, we do see a reduction in INP concentrations due to the SP2 laser. Thus, INPs internally mixed with rBC generally cause overestimations of the rBC contribution to INP concentrations in the SP2-CFDC. To account for this, we determined the fraction of "attached-type" particles (section 2.3). Interestingly, we find that the attached-type fraction correlates well with the number fraction of deactivated INP for the internally mixed NX-illite case. The INEs in NX-illite are refractory, and therefore will not be fully vaporized when an attached rBC particle is heated to 4000 K; thus, scattering material would traverse SP2 laser and these particles would appear as "attached" in the SP2 analysis. From Figure 8, we see that 97% of the attached-type fraction was deactivated after exposure to the SP2 laser. This is consistent with results from Levin et al. [2014], who used a different analysis to estimate the number of attached-type particles, and found that, at minimum, 74% of mixed Aquadag®-ATD particles were deactivated as INP following exposure to the SP2 laser. Thus, for refractory INE attached to rBC, the attached-type fraction can be used to estimate the number of deactivated, rBC-containing INP.

In contrast, the attached-type fraction does not correlate with the number of deactivated INP for the internally mixed SRFA and Snomax® cases. Here, unlike NX-illite, the INEs from SRFA and Snomax® are non-refractory/heat-labile. Thus, when rBC heats to 4000 K in the SP2 laser, any attached, non-refractory INEs are completely evaporated or destroyed. To confirm this, we estimated the fraction of rBC-SRFA/Snomax® particles that were affected by the SP2-laser. During one-pot nebulization, or what we are calling the "internally mixed" scenario, some fraction of particles will be pure rBC, some fraction will be pure SRFA/Snomax®, and a final fraction will be truly internally mixed. As a conservative estimate, we assume that all particles that contain incandescing material also contain INPs. From the SP2 raw data, 96 and 76% of all Aquadag®-SRFA and Aquadag®-Snomax® particles in the "internally mixed" scenario contained incandescent material, respectively. Thus, only 96 and 76% of the particles could contain rBC that is both greater than 90 nm and physically attached to an INE. From Figure 8, the fraction of deactivated INP was 90% for "internally mixed" Aquadag®-SRFA and 69% for "internally mixed" Aquadag®-Snomax® particles. Thus, 94% of the incandescent Aquadag®-SRFA particles and 91% of the incandescent Aquadag®-Snomax® particles were deactivated following exposure to the SP2 laser. From the combined above analyses, we believe that heating rBC to its vaporization temperature of ~4000 K will destroy >90% of any physically attached INE. Therefore, while this technique cannot tell us the number of rBC INP, it can tell us the number of rBC-containing INP. Thus, we recommend using the terminology "rBC-containing contribution," instead of the previously used "rBC contribution."

**4 Conclusions and Recommendations for the Future**

We determined the effect of the SP2 laser on the ice nucleation efficiency of Snomax®, NX-illite, and SRFA internally and externally mixed with Aquadag®. In addition, we quantified how both SP2 absolute laser power and sheath-to-sample ratios affected the physical properties of rBC-SP2 exhaust. Consequently, we have bounded the SP2 conditions under which rBC particles are fully vaporized. The results of this study, which are generally applicable to all SP2 instruments, are as follows: (a) the minimum absolute power to fully vaporize Aquadag® particles > 125 nm (>0.76 fg) is 930 nW/(220-nm PSL); (b) new particles are formed in the SP2 exhaust from vaporized Aquadag® particles *via* homogeneous nucleation, and are dependent on the initial rBC particle mass; (c) the size distributions and ice nucleating efficiencies of proxies of biological particles, illite-rich dust, and BrC are minimally affected by the SP2 laser; and (d) external mixtures of rBC with INP proxies minimally affect their ice nucleation efficiency, but internal mixtures can cause overestimations of the rBC contribution to INP concentrations.

The absolute power needed to fully vaporize Aquadag® particles was calculated to be 930 nW/(220-nm PSL). This is more than twice the power recommended to achieve unity detection of a 0.7 fg particle and over three-times the minimum absolute laser power needed to achieve quantitative measurements of rBC mass in the accumulation mode (Schwarz et al., 2010). Previous studies indicate that rBC particles that undergo laser induced-incandescence at much lower laser powers pass through the laser only partially ablated (Michelsen et al., 2007). We only found evidence of partially ablated particles at lower absolute laser powers. With sufficient absolute laser power, we affirmed that rBC particles are vaporized in the SP2 cavity via laser-induced incandescence and that new particles homogeneously nucleate from the vapors by taking TEM images of Aquadag® SP2 exhaust with the laser on and off. When the laser is off, the particles are crumpled sheets of graphitic carbon both spheroid in shape and approximately 500 nm in diameter; in contrast, when the laser is on, the same field of view is dominated by ~10 nm spherical monomers and ~20-30 nm agglomerates. High resolution images of these particles show these monomers do not exhibit any structural motifs, including the repetitive spacing indicative of graphitic carbon. Thus, we conclude that the exhaust of the SP2, when sampling Aquadag® with sufficient laser power , consists of small, amorphous particles formed from the homogeneous nucleation of vaporized rBC.

Finally, the SP2 laser has no discernable effect on the original size distribution or the ice nucleation efficiency of Snomax®, NX-illite, or SRFA. This was also true for external mixtures of these INP proxies with rBC, suggesting that ice nucleating proteinaceous material, atmospheric mineral dust, and atmospheric BrC will be minimally affected by this technique. When these proxies were internally mixed with rBC, however, reductions in INP concentrations were found. Thus, a high-degree of internal mixing may lead to an artificial overestimate the contribution of rBC particles to INP concentrations.

To account for all aforementioned results, the following guidelines have been adopted for operation of the CSU SP2 for its use as a prefilter to additional aerosol instruments, and are recommended for the community who wish to use the SP2 as a prefilter to filter out rBC particles >0.76 fg.

(1) Prior to conducting pre-filter experiments, assess the absolute laser power using PSL calibrations.

(2) If complete vaporization of >0.76 fg particles is desired, adopt a minimum laser power of 930 nW/(220-nm PSL). Note that this laser power is temperature dependent.

(3) To avoid contaminating the optics when sampling the SP2 exhaust with an additional instrument, it is recommended that the SP2 first be operated with a filter on the inlet to ensure that the cavity is particle-free. Additional flow from the aerosol instrument sampling the exhaust should be added to the exhaust line prior to isolating the SP2 exhaust pump. This is best achieved with a dedicated system to facilitate this switch (Section 2.8).

(4) The term "rBC-containing contribution" should be used to clarify that the ice nucleating properties of particles internally mixed with rBC can be affected by the SP2 laser, and that the contributions of these particles to INP number concentrations are not easily deconvoluted.

The results presented here bolster confidence in the use of the SP2 as a pre-filter for rBC to other aerosol instruments. This is especially true for online ice nucleation instruments, where the ice nucleation behavior of ambient rBC particles is poorly constrained. With an adequate laser intensity, rBC particles >90-nm, which are the most likely to nucleate ice, can be effectively removed from aerosol populations. This will allow for the direct determination of the contribution of rBC-containing particles to INP concentrations from real world samplings such as prescribed burns and wildfires.

## Acknowledgments and Data

This material is based upon work supported by the NSF Division of Atmospheric and Geospace Sciences under award number 1433517. Funding for PJD, EJTL, and SMK and other logistical support was provided by the NASA Earth Science Division under award NNX12AH17G. The data used are listed in the references, tables, supplements and in a digital repository at CSU (https://dspace.library.colostate.edu).

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

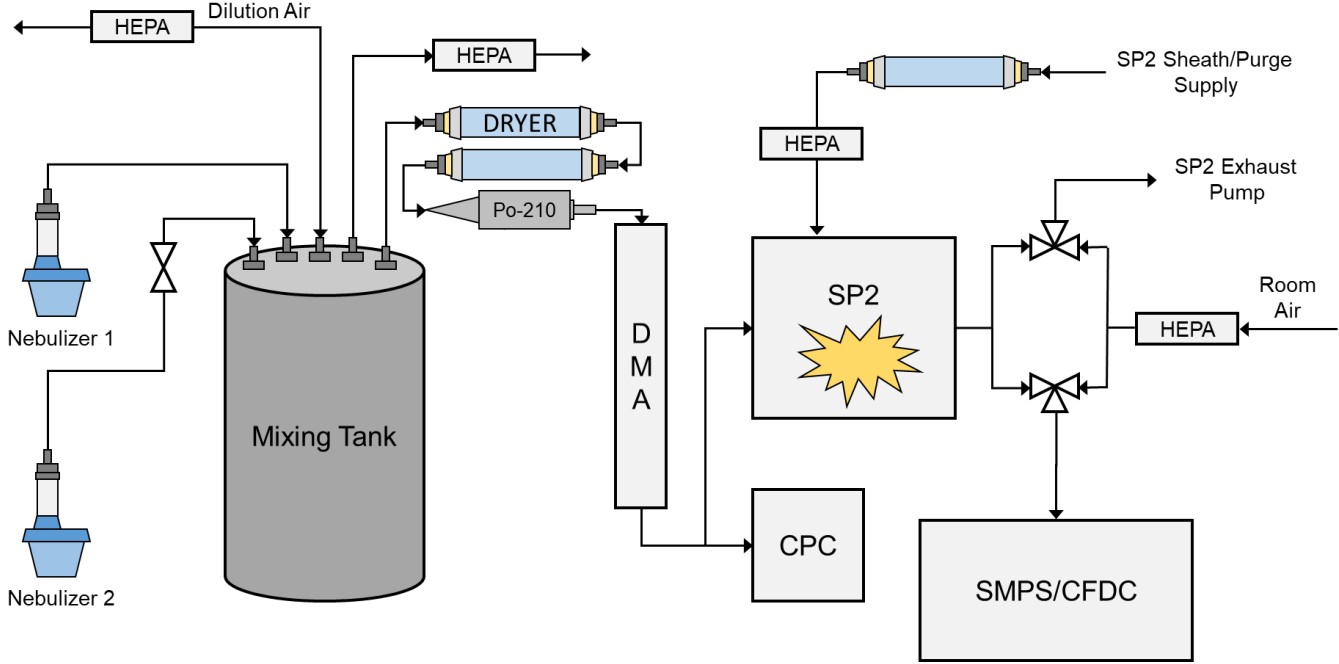

**Figure 1. The experimental setup for unmixed, internal mixing, and external mixing studies with the SP2-SMPS/CFDC.**

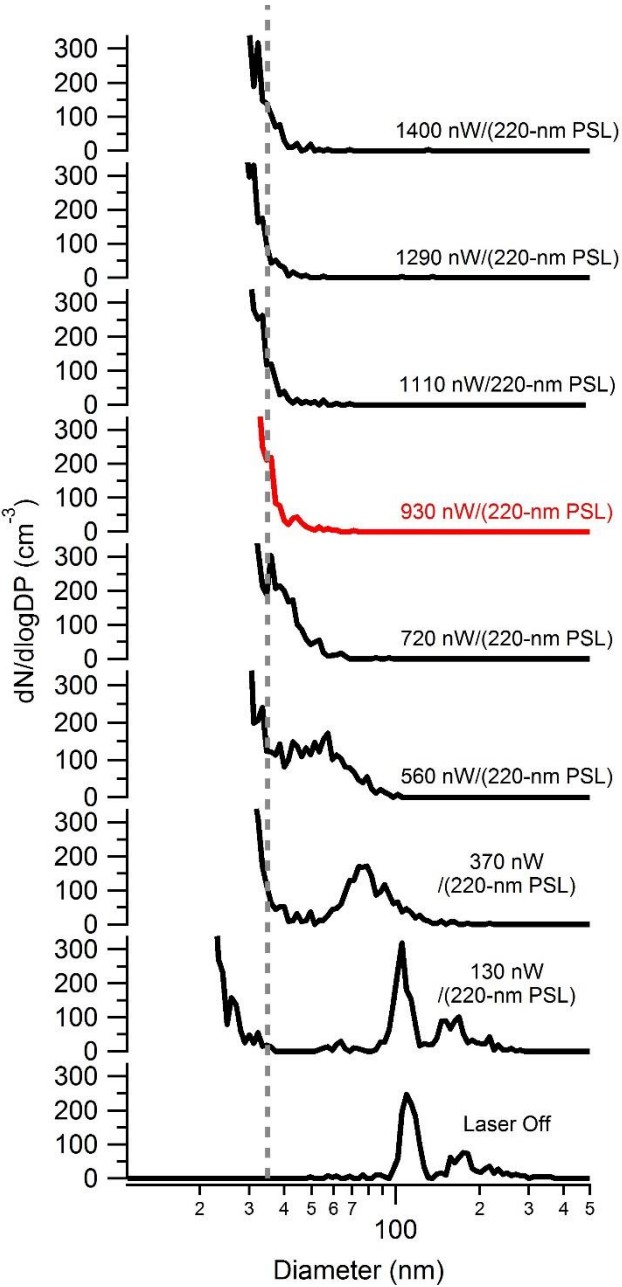

**Figure 2: The size distribution of 125-nm Aquadag® SP2 exhaust as a function of SP2 absolute laser power. As shown, the original Aquadag® particle is only partially ablated until 930 nW/(220-nm PSL) (red curve), where is it fully vaporized.**

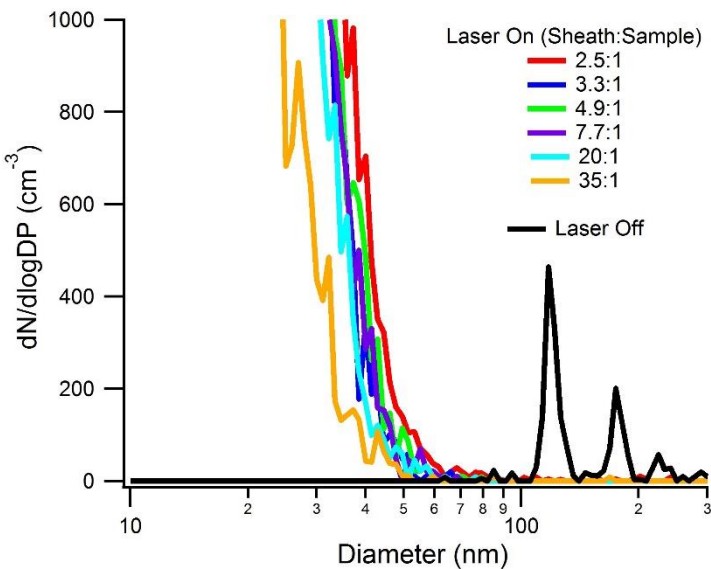

**Figure 3: Size distribution of the SP2 exhaust from 125-nm Aquadag® particles at a concentration of 600 cm$^{-3}$ under different sheath-to-sample ratios. For reference, the size distribution of a 125-nm Aquadag® particle at a sheath-to-sample ratio of 4.9:1 is shown in black. The absolute SP2 laser power was 1290 nW/(220-nm PSL).**

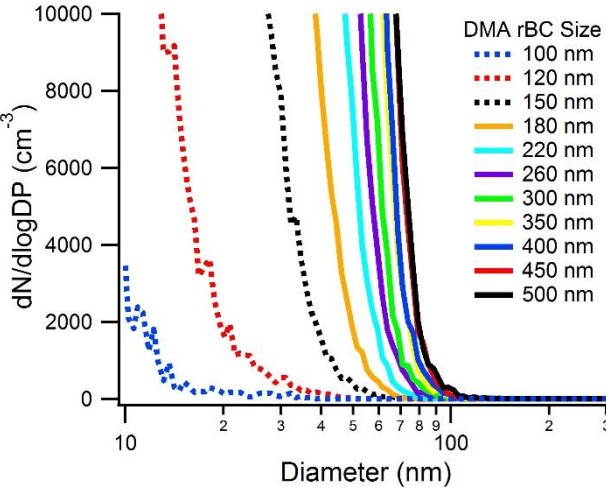

**Figure 4: Size distribution of SP2 exhaust from different sized-selected rBC particles. The concentration of size-selected rBC particles was approximately 1000 cm⁻³ and the sheath:sample was 4.9:1.**

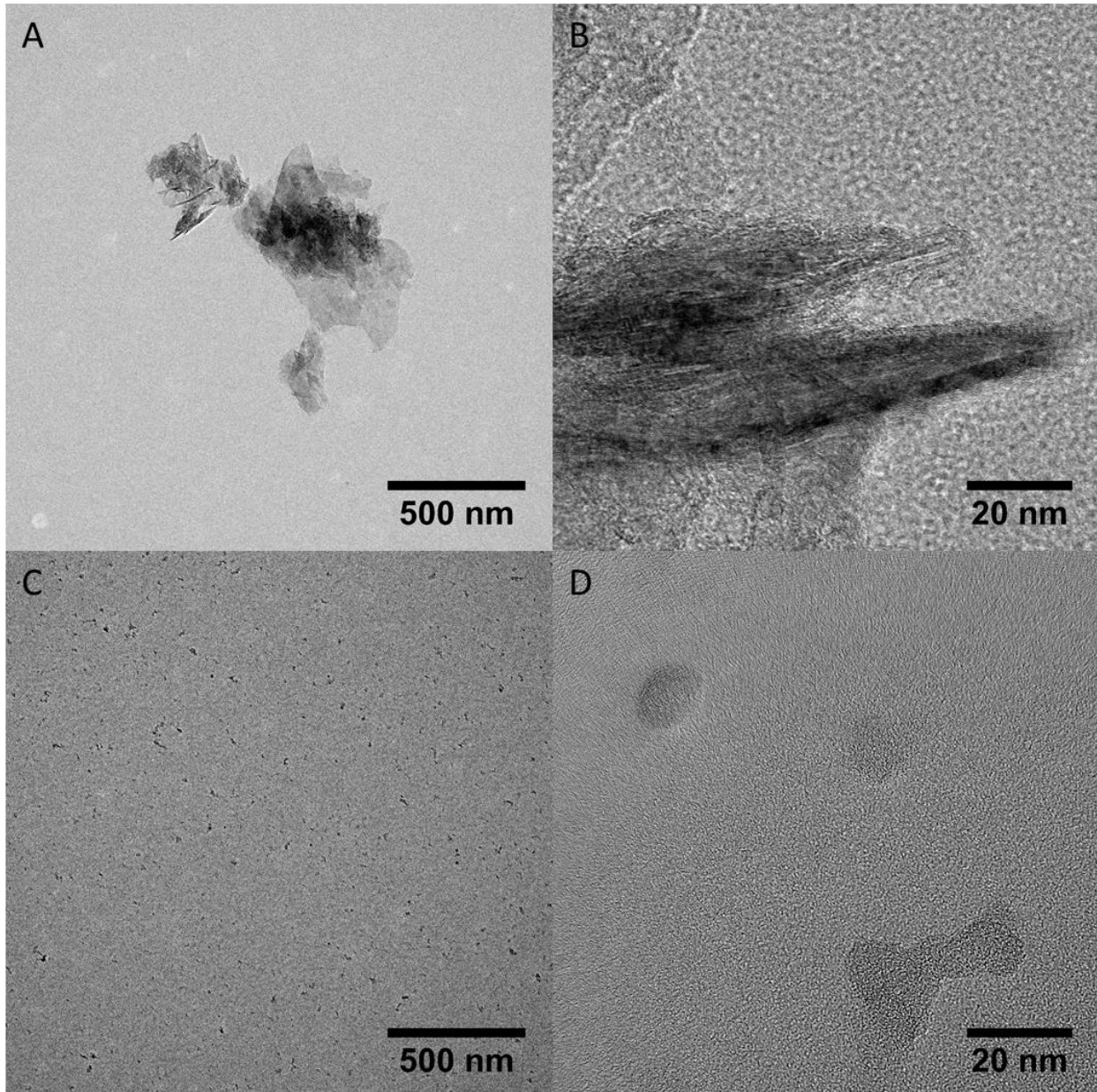

**Figure 5: TEM images of Aqudag® with the SP2 laser off (A, B) and on (C, D). The higher resolution images show that the new particles do not exhibit structural motifs indicative of graphitic carbon. Thus, the new particles are likely formed from homogeneous nucleation of rBC vapors instead of fragments of the original particles.**

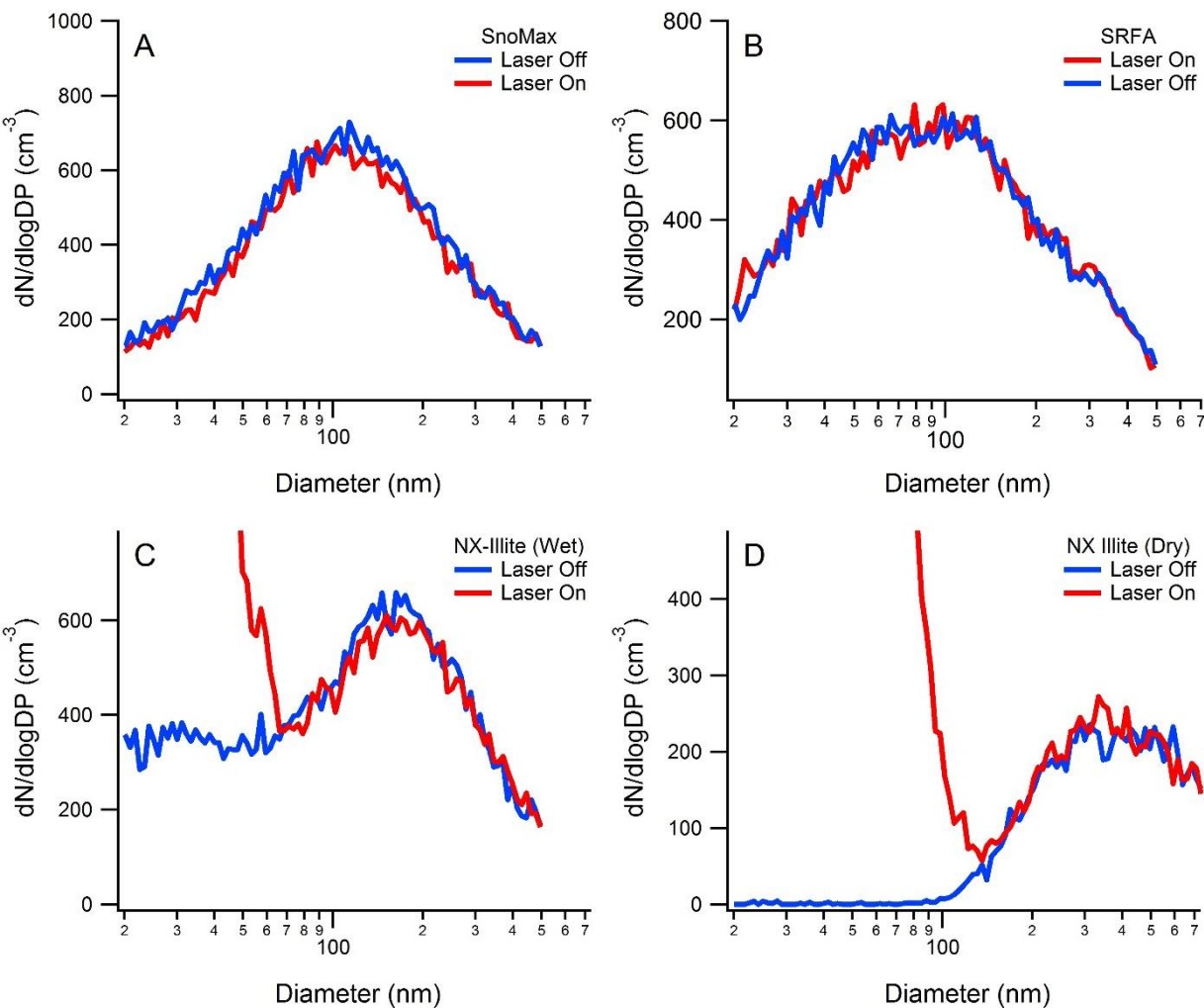

**Figure 6: The size distribution of Snomax® (A), SRFA (B), NX-illite wet generated (C), and NX-illite dry generated (D). The size distributions were measured when the SP2 laser was both on and off. As shown, both Snomax® and SRFA are unaffected by the SP2 laser. A new particle peak does appear for NX-illite when the SP2 laser is on, but the original size distribution is minimally affected.**

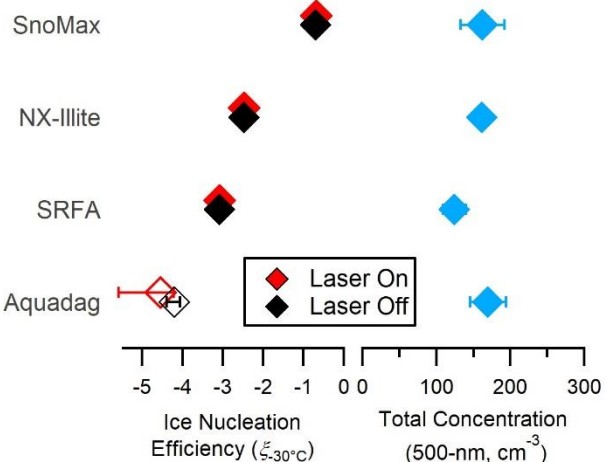

**Figure 7. The ice nucleation efficiencies and total concentrations of pure, 500-nm size-selected Snomax®, NX-illite, SRFA, and Aquadag®. Filled and unfilled symbols are above and below the detection limit of the CFDC when corrected for SP2 dilution, respectively. The errors bars represent one standard deviation of the mean from at least three measurement periods.**

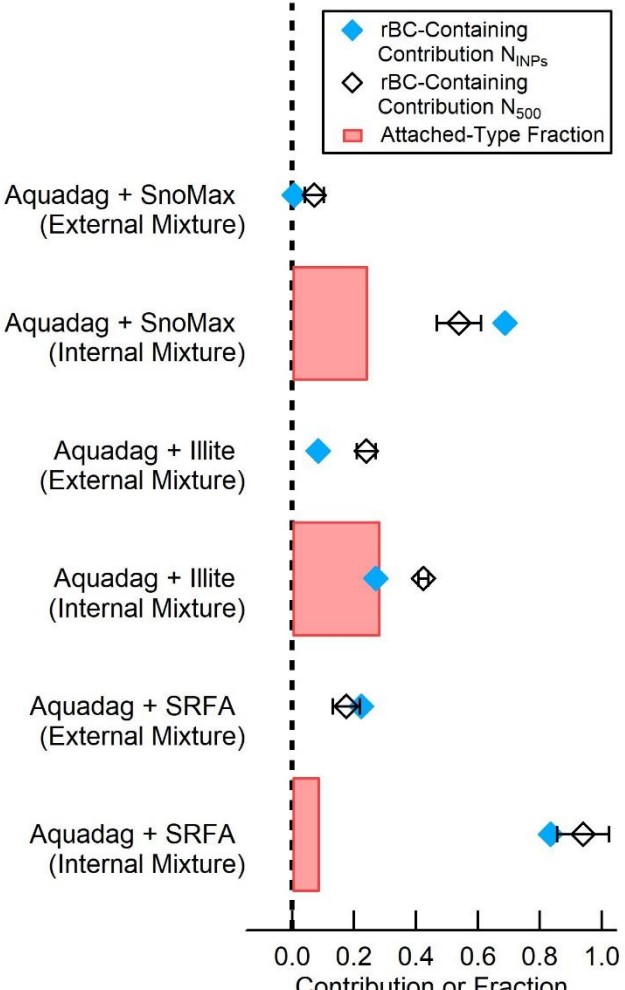

**Figure 8. The rBC-containing contribution of INPs ($N_{INPs}$) active at -30 °C and 105% RH, particles greater than 500 nm**

5  **($N_{500}$), and the number fraction of attached-type particles for the internally mixed scenarios. As shown, the attached-type**

**fraction correlates poorly with rBC contributions to $N_{INPs}$. The errors bars represent one standard deviation of the mean**

**from at least three measurement periods.**