# Peer review of "Use of the Single Particle Soot Photometer (SP2) as a pre-filter for ice nucleation measurements: Effect of particle mixing state and determination of SP2 conditions to fully vaporize refractory black carbon"

_Atmospheric Measurement Techniques, 2017_

## Referee Comment (RC1) · Anonymous Referee #1 · 9 Jan 2018

\* General comments

The authors report on a set of laboratory measurements to investigate the use of a single particle soot photometer (SP2) as a pre-filter for ice nucleation measurments, in order to determine the concentration of rBC-containing particles that act as ice nucleating particles (INPs). This technique has been applied by some of the authors in previous studies. The present study extends on this work by investigating the condi-

tions required to fully vaporize Aquadag BC in an SP2, and by examining the effect of the SP2 laser on internal and external mixtures of INP proxies and Aquadag. The topic of the paper is interesting and suitable for AMT. Asides from the relevance of INPs, the results are also of interest to Laser-Induced Incandescence studies. The experimental design and measurements are both good quality. For the most part the manuscript is well-written and the results well presented. However, I have a number of comments that I think must be addressed before the manuscript is considered for publication in AMT.

One of the most important results I learned from this study is that the technique in question can't be used on internal mixtures of INPs and BC. The SP2 laser deactivates some of the INPs and it is not simple to work out exactly how much. This is a somewhat negative result as it severly limits the applicability of the technique. But it is still an important point that I believe requires more discussion. Most ambient BC samples contain a sizeable fraction of internally mixed BC. Is my conclusion premature, and do the authors think the technique can still be used to determine the concentration of rBC-containing particles that act as INPs in these situations, albeit with an increased measurement uncertainty? If not, under what conditions can the technique be safely applied?

What I presume to be a smaller peak of doubly-charged particles appears below 200 nm in Figs. 2 and 3. I would expect it to show up at 250 nm. What's going on here?

A number of results are concluded from the rBC-containing contribution to N_INPs values shown in Fig. 8. I found these results quite difficult to comprehend within this framework. Perhaps this is just me, but I would encourage the authors to think about how they could more effectively communicate their main results. One suggestion for the pure proxies result is given below in my specific comments. I would also suggest the authors at least write out somewhere in the manuscript how the rBC-containing contribution to N_INPs is calculated, to facilitate understanding of the discussions based around this parameter.

* Specific comments

P1, L20: 'Fluence' should be changed to 'power'. Fluence is energy per area, and a power value is given in the square brackets.

P2, L21: 'Direct' repeated in the final part of this sentence.

P5, L26: What were the typcial concentrations in the tank, to indicate timescale of coagulation.

P6, L13: Extra 'is then' in this sentence.

P7, L3: Minor point but this comparison does not hold up exactly. In pulsed LII, particles are generally heated to below their vaporisation temperature by choice, to ensure they don't lose mass due to evaporation. To account for this, particle temperatures are typically monitored by 2-color pyrometry, and thus the particles are not required to have 'uniform detection efficiency'. A more recent and relevant reference for pulsed LII might be Michelsen et al., 2015.

P7, L22: What sheath:sample air flow rate ratio was used for these measurements.

P8, L6: Could sintering of Aquadag fragmnets possibly also cause the change in microstructure from graphitic to amorphous?

P8, L9: ATD has not been defined previously in the manuscript.

P8, L35: 0.002% of what? The total particle mass?

P9, L10: Does Fig. 7 show CFDC measurements for when the SP2 laser was on or off? To more easily show that the SP2 laser minimally affected the ice nucleation ability of the pure proxies, would it be possible to plot both SP2 laser on and off measurements on this Fig.? I believe this would be easier to understand than the way this point is currently made, which is through the rBC containing contributions to N_INPs of the pure proxies shown in Fig. 8.

[Figure]

P9, L21: Missing word between 'determined' and 'for'. E.g. '...previously determined value for ...'

P9, L38: This conclusion is stated too strongly. Only 2 of the 3 cases show the laser did not affect INP efficacy - I rather consider the SRFA case as an example when it did affect it - and only a limited range of experiments have been conducted (1 type of incandescing particle, 1 external mixture of each type).

P10, L2: Please provide some justification for this 30% threshold. I don't imagine this represents the random uncertainty in taking the difference between two low INP concentration measurements, since one might also obtain negative fractions of similar magnitude, and the pure proxy measurements show this doesn't seem to be the case.

P10, L5: Could it also be the case that the attached-type particle measure introduced in section 2.3 is not sensitive enough? Since this measure does not seem to have been validated against indepedent measurements or Moteki's method I don't think this can be ruled out.

P10, L7: Taking this point even further, can it be that since the SRFA was coating the Aquadag and not just attached to it, the SRFA material evaporated more or less completely, removing all possible INPs. A possible suggestion for a future experiment could be given: comparison of the rBC-containing contributions of internal mixtures generated by the current mixed-solution method and by coagulating the two particle types.

P10, L9: Typo here 'INE' instead of 'INP'. Also, this sentence seems to say exactly the same thing as the one before it.

P10, L36: Care must be taken not to over-generalize here. Only Aquadag particles have been investigated in this study. Other types of BC will absorb varying amounts of energy from the SP2 laser based on their imaginary refractive index, and it is not clear homogenous nucleation will always result from vaporized BC.

[Figure]

P11, L23: The big caveat here is that this determination will not work if a sizeable fraction of INPs are internally mixed with BC, which is the case for burns and wildfires. Therefore, I do not agree with this statement.

P17, Fig. 3: Please indicate in the caption that particle concentrations were kept constant at 600 cm-3

P18, Fig. 4: Please indicate in the caption what sheath:sample air flow rate ratio was used for these measurements.

P21 and 22, Figs. 7 and 8: Please provide an explanation here or in the main text of exactly what the error bars represent.

P22, Fig. 8: X-axis label missing (even if one can still kind of figure out what is shown).

* References

Michelsen, H. A., Schulz, C., Smallwood, G. J. and Will, S.: Laser-induced incandescence: Particulate diagnostics for combustion, atmospheric, and industrial applications, Progress in Energy and Combustion Science, 51, 2–48, doi:10.1016/j.pecs.2015.07.001, 2015.

---

## Referee Comment (RC2) · Anonymous Referee #2 · 8 Mar 2018

This paper presents a characterisation of an SP2 instrument that is to be used as a pre-filter for ice nucleation experiments. It presents a series of careful experiments to examine the evaporation of rBC by the incandescence laser as a function of laser power and shows that even at low laser powers an ultrafine particle population is observed in the exhaust of the instrument. TEM analysis shows that this is due to nucleation and not fragmentation. The conclusion is that a laser power of 930 nW is required for complete vaporization. The effect of black carbon on the ice nucleation of a range of standard

proxies was carried out and it is shown that only when black carbon is internally mixed with the proxy does its IN efficiency change. These conclusions therefore allow the effectiveness of black carbon particles as IN to be separated from other IN as long as the population is externally mixed. Where significant internal mixing has taken place then its use as a probe may be less unequivocal. This characterisation is important and is certainly worthy of publication in AMT. The paper concludes that the results bolster confidence in the method to separate the effects of rBC in IN experiments and cites wildfires as an important area. I would question whether the tone should be so optimistic since significant dust is often present in wildfires and the mixing of rBC and dust in the near field of fires is not well characterised. My interpretation of the results presented here is that this is a potential shortcoming that cannot easily be overlooked nor tackled when using ambient data. This should be included in the final discussion and ant methods to identify its influence identified. The paper is largely very well written in my view and the figures are clear and understandable. There are one or two places where the text could be made more readable and these are identified below. It might be worth commenting in the discussion on whether the nucleation arises from the organic matter evaporating and then re-nucleating as well as the core. Whilst this is not important in your experiment since the particles were nearly all composed of rBC but for particles with significant coating, such as biomass burning particles it could a much bigger effect. One might also expect significant condensational growth of the nucleated particles in these conditions. Given the conclusions are focused on the use of the instrument for investigating the IN effectiveness of biomass burning this is worth including. Page 3 line 36: This line seems to alternate between singular and plural (finesses/face) Page 4 lines 5-6: s in figure and S in text Page 4 line 25: The description of the positions in not clear. Is this the time delay from the first point the signal passes the reference threshold? If so, which detector? This needs to be clear. Page 4: line 28: depend(a)nt dependent Page 6 line 7: A lesson learned from bitter experience I suspect Page 7 line 4: "of at least At" Figure 3: caption state the laser power used Page 9 lines 24-27: I do not feel that "rBC containing contribution" in figure

8 is well described at all. What is "the effect" on laser power? and what does the scale on the x axis of the figure represent? This needs clarification. Page 10 line 16 of (the) SP2 laser Page 10 line 26: particles

———————————————————

---

## Author Comment (AC1) · 5 Apr 2018

The reviewer's comments are bolded and italicized while our comments are in plain text.

Response to Reviewer #1

**\* General comments**

The authors report on a set of laboratory measurements to investigate the use of a single particle soot photometer (SP2) as a pre-filter for ice nucleation measurments, in order to determine the concentration of rBC-containing particles that act as ice nucleating particles (INPs). This technique has been applied by some of the authors in previous studies. The present study extends on this work by investigating the conditions required to fully vaporize Aquadag BC in an SP2, and by examining the effect of the SP2 laser on internal and external mixtures of INP proxies and Aquadag. The topic of the paper is interesting and suitable for AMT. Asides from the relevance of INPs, the results are also of interest to Laser-Induced Incandescence studies. The experimental design and measurements are both good quality. For the most part the manuscript is well-written and the results well presented. However, I have a number of comments that I think must be addressed before the manuscript is considered for publication in AMT.

The authors would first like to thank Reviewer #1 for his/her insightful comments, which have improved the clarity and utility of this manuscript.

One of the most important results I learned from this study is that the technique in question can't be used on internal mixtures of INPs and BC. The SP2 laser deactivates some of the INPs and it is not simple to work out exactly how much. This is a somewhat negative result as it severly limits the applicability of the technique. But it is still an important point that I believe requires more discussion. Most ambient BC samples contain a sizeable fraction of internally mixed BC. Is my conclusion premature, and do the authors think the technique can still be used to determine the concentration of rBC-containing particles that act as INPs in these situations, albeit with an increased measurement uncertainty? If not, under what conditions can the technique be safely applied?

The authors agree that this is one of the most important results which we have learned both from this study and the previously published manuscript by Levin et al. (2014). The authors, however, do not believe that this result severely limits the applicability of this technique. We concede that our initial discussion about the attached-type analysis was ambiguous, *i.e.*, it did not state clearly why our quantification of attached-type particles did not predict the rBC-containing contribution. It appears that the reviewer interpreted the results to mean that an unknown fraction of internally mixed particles is affected by the laser. We suggest, and the it seems the reviewer agrees due to their specific comments on Page 10, Lines 5 and 7, that the attached-type analysis does not predict the number of rBC-containing INP for SRFA and Snomax® because their ice nucleating entities (INEs) are non-refractory, *i.e.*, as the attached rBC particle is heated to 4000 K, the INEs from SRFA and Snomax® are fully vaporized. Thus, these particles will not appear as an "attached-type" in our analysis, since no scattering material will survive the SP2 laser. During one-pot nebulization, or what we are calling the "internally mixed" scenario, some fraction of

particles will be pure rBC, some fraction will be pure SRFA/Snomax®, and a final fraction will be truly internally mixed. For SRFA and Snomax®, we have conservatively estimated the number of truly internally mixed particles, and find that >90% of those particles' ice nucleating ability were affected by the SP2 laser.

Additionally, NX-illite INEs are refractory, and therefore will not fully vaporize when heated to 4000 K. Thus, the attached-type fraction is analogous to the fraction of particles that are truly internally mixed. Figure 7, we see that 97% of the attached type-particles were deactivated due to exposure to the SP2 laser. Thus, the authors believe that the SP2 laser affects the ice nucleation ability of a large fraction of INPs internally mixed with rBC. These results suggest that the term "rBC-containing contribution," which is still useful to the ice nucleation community, is valid with a possible overestimation of <10%. We have clarified this in the manuscript by amending the final paragraph of Section 3.4 to now read:

"When the INP proxies are internally mixed with rBC, we do see a reduction in INP concentrations due to the SP2 laser. Thus, INPs internally mixed with rBC generally cause overestimations of the rBC contribution to INP concentrations in the SP2-CFDC. To account for this, we determined the fraction of "attached-type" particles (section 2.3). Interestingly, we find that the attached-type fraction correlates well with the number fraction of deactivated INP for the internally mixed NX-illite case. The INEs in NX-illite are refractory, and therefore will not be fully vaporized when an attached rBC particle is heated to 4000 K; thus, scattering material would traverse SP2 laser and these particles would appear as "attached" in the SP2 analysis. From Figure 8, we see that 97% of the attached-type fraction was deactivated after exposure to the SP2 laser. This is consistent with results from Levin et al. [2014], who used a different analysis to estimate the number of attached-type particles, and found that, at minimum, 74% of mixed Aquadag®-ATD particles were deactivated as INP following exposure to the SP2 laser. Thus, for refractory INE attached to rBC, the attached-type fraction can be used to estimate the number of deactivated, rBC-containing INP.

In contrast, the attached-type fraction does not correlate with the number of deactivated INP for the internally mixed SRFA and Snomax® cases. Here, unlike NX-illite, the INEs from SRFA and Snomax® are non-refractory/heat-labile. Thus, when rBC heats to 4000 K in the SP2 laser, any attached, non-refractory INEs are completely evaporated or destroyed. To confirm this, we estimated the fraction of rBC-SRFA/Snomax® particles that were affected by the SP2-laser. During one-pot nebulization, or what we are calling the "internally mixed" scenario, some fraction of particles will be pure rBC, some fraction will be pure SRFA/Snomax®, and a final fraction will be truly internally mixed. As a conservative estimate, we assume that all particles that contain incandescing material also contain INPs. From the SP2 raw data, 96 and 76% of all Aquadag®-SRFA and Aquadag®-Snomax® particles in the "internally mixed" scenario contained incandescent material, respectively. Thus, only 96 and 76% of the particles could contain rBC that is both greater than 90 nm and physically attached to an INE. From Figure 8, the fraction of deactivated INP was 90% for "internally mixed" Aquadag®-SRFA and 69% for "internally mixed" Aquadag®-Snomax® particles. Thus, 94% of the incandescent Aquadag®-SRFA particles and 91% of the incandescent Aquadag®-Snomax® particles were deactivated following exposure to the SP2 laser. From the combined above analyses, we believe that heating rBC to its vaporization temperature of ~4000 K will destroy >90% of any physically attached

INE. Therefore, while this technique cannot tell us the number of rBC INP, it can tell us the number of rBC-containing INP. Thus, we recommend using the terminology "rBC-containing contribution," instead of the previously used "rBC contribution."

**What I presume to be a smaller peak of doubly-charged particles appeamrs below 200 nm in Figs. 2 and 3. I would expect it to show up at 250 nm. What's going on here?**

The TSI DMA selects monodisperse particles based on their electrical mobility. The electrical mobility of a particle is affected both by its diameter-to-charge ratio, but also the Cunningham slip correction factor—which itself is dependent on diameter (DeCarlo et al., 2004). Thus, when using a DMA to size select 125-nm particles, we expect doubly charged particles of the same electrical mobility to be 192-nm in diameter.

A number of results are concluded from the rBC-containing contribution to  $N_INPs$  values shown in Fig. 8. I found these results quite difficult to comprehend within this framework. Perhaps this is just me, but I would encourage the authors to think about how they could more effectively communicate their main results. One suggestion for the pure proxies result is given below in my specific comments. I would also suggest the authors at least write out somewhere in the manuscript how the rBC-containing contribution to  $N_INPs$  is calculated, to facilitate understanding of the discussions based around this parameter.

Upon review, the authors agree that the results in Figure 8 are difficult to comprehend because there are several key pieces of information. As suggested by the reviewer, we have removed the pure proxies from Figure 8, and added the laser's effect on their ice nucleation properties into Figure 7. We now feel that both Figures 7 and 8 effectively communicate the main results. The authors thank the reviewer for their suggestion.

Furthermore, the following description of how we calculate the rBC-containing contribution has been added Section 2.4:

"and determined the rBC contribution. The rBC contribution is defined as:

$$rBC \text{ Contribution} = 1 - \frac{(N_{INP,SP2 \ Laser \ On})/(N_{Total})}{(N_{INP,SP2 \ Laser \ Off})/(N_{Total})}$$
(3)

where  $N_{INP, SP2 Laser On}$  is the number or INP measured by the CFDC and corrected for dilution when the sampling the SP2 exhaust with the laser on,  $N_{INP, SP2 Laser Off}$  is the number or INP measured by the CFDC and corrected for dilution when the sampling the SP2 exhaust with the laser off, and  $N_{Total}$  is the total number concentration of aerosol measured by the CPC upstream of the SP2."

**\* Specific comments P1, L20: 'Fluence' should be changed to 'power'. Fluence is energy per area, and a power value is given in the square brackets.**

All instances of "fluence" have now been changed to "power."

**P2, L21: 'Direct' repeated in the final part of this sentence.**

Corrected. Thank you.

**P5, L26: What were the typcial concentrations in the tank, to indicate timescale of coagulation.**

Thank you. The last two sentences of Section 2.7 now read:

"While coagulation inside the mixing tank is possible, it is assumed that the concentration of coagulated particles is small. Typical concentrations during the external mixing scenario were  $\sim$ 100-200 cm-3 of 500-nm size-selected particles."

**P6, L13: Extra 'is then' in this sentence.**

Corrected. Thank you.

P7, L3: Minor point but this comparison does not hold up exactly. In pulsed LII, particles are generally heated to below their vaporisation temperature by choice, to ensure they don't lose mass due to evaporation. To account for this, particle temperatures are typically monitored by 2-color pyrometry, and thus the particles are not required to have 'uniform detection efficiency'. A more recent and relevant reference for pulsed LII might be Michelsen et al., 2015.

This was noted and the word "fully" was removed from this clause. Additionally, the Michelson et al., 2015 paper was added to the reference section.

**P7, L22: What sheath:sample air flow rate ratio was used for these measurements.**

In Figure 4, the sheath:sample air flow rate ratio was 4.9:1. The following clause was added to Page 7, Line 22 and the caption for Figure 4:

"... and the sheath:sample was 4.9:1."

**P8, L6: Could sintering of Aquadag fragmnets possibly also cause the change in microstructure from graphitic to amorphous?**

Since sintering, by definition, does not involve a solid-to-liquid phase change, the authors do not believe that the particles would change their microstructure from graphitic to amorphous.

**P8, L9: ATD has not been defined previously in the manuscript.**

Thank you. The first instance of ATD has now been changed to "Arizona Test Dust (ATD)"

**P8, L35: 0.002% of what? The total particle mass?**

This line has been amended to state "0.002% of all particles by number."

P9, L10: Does Fig. 7 show CFDC measurements for when the SP2 laser was on or off? To more easily show that the SP2 laser minimally affected the ice nucleation ability of the pure proxies, would it be possible to plot both SP2 laser on and off measurements on this Fig.? I believe this would be easier to understand than the way this point is currently made, which is through the rBC containing contributions to N\_INPs of the pure proxies shown in Fig. 8.

See text in "\*General Comments" section.

*P9, L21: Missing word between 'determined' and 'for'. E.g. '...previously determined value for ...'*

Corrected. Thank you.

**P9, L38: This conclusion is stated too strongly. Only 2 of the 3 cases show the laser did not affect INP efficacy - I rather consider the SRFA case as an example when it did affect it - and only a limited range of experiments have been conducted (1 type of incandescing particle, 1 external mixture of each type).**

Agreed. We have added a possible explanation for the behavior of externally mixed rBC and SRFA and several hedging words to the sentence in question. The last two sentences of that paragraph now read"

"It may be possible that liquid SRFA particles have a higher propensity to coagulate in the mixing tank with rBC particles than solid Snomax® or NX-illite particles; however, we have no evidence to corroborate this statement. Nonetheless, we find that incandescing particles generally do not affect the efficacy of mineral dust and biological INP proxies when they are not physically attached.

**P10, L2: Please provide some justification for this 30% threshold. I don't imagine this represents the random uncertainty in taking the difference between two low INP concentration measurements, since one might also obtain negative fractions of similar magnitude, and the pure proxy measurements show this doesn't seem to be the case.**

It seems that reviewer interpreted "30% or greater" to be a general statement. It was the authors' intent to merely point out that the rBC-containing contribution from the three internal-mixture cases in this study were 30% or greater. This number will fluctuate depending on the efficiency

of the INP proxies as well as the degree of mixing. Thus, we have deleted the phrase "of 30% or greater" from this sentence.

**P10, L5: Could it also be the case that the attached-type particle measure introduced in section 2.3 is not sensitive enough? Since this measure does not seem to have been validated against indepedent measurements or Moteki's method I don't think this can be ruled out.**

P10, L7: Taking this point even further, can it be that since the SRFA was coating the Aquadag and not just attached to it, the SRFA material evaporated more or less completely, removing all possible INPs. A possible suggestion for a future experiment could be given: comparison of the rBC-containing contributions of internal mixtures generated by the current mixed-solution method and by coagulating the two particle types.

The authors agree with the reviewer that "the SRFA material evaporated more or less completely, removing all possible INPs." A similar statement was made in the original text, Page 10, Line 9:

"Thus, in the SP2, the INEs are found in a non-refractory coating on the rBC core."

The following sentence has been added to make this inference clearer to the reader:

"Here, unlike NX-illite, the INEs from SRFA and Snomax® are non-refractory/heat-labile. Thus, when rBC heats to 4000 K in the SP2 laser, any attached, non-refractory INEs are completely evaporated or destroyed."

**P10, L9: Typo here 'INE' instead of 'INP'. Also, this sentence seems to say exactly the same thing as the one before it.**

The authors appreciate the reviewers careful reading for typos, however, the use of INE was intentional and was defined on Page 9, Line 17.

**P10, L36: Care must be taken not to over-generalize here. Only Aquadag particles have been investigated in this study. Other types of BC will absorb varying amounts of energy from the SP2 laser based on their imaginary refractive index, and it is not clear homogenous nucleation will always result from vaporized BC.**

While it is true that Aquadag®, and not ambient BC, was used in this study, we consider its vaporization behavior in the SP2 cavity to behave like ambient BC as it has a similar vaporization temperature and it is used a standard calibration material for the SP2 (Baumgardner et al., 2012; Moteki & Kondo, 2010). Additionally, we also explored the size distribution of ambient air containing rBC and found similar behavior at low sizes in the size distribution (Figure S4). Nonetheless, we have amended the sentence in question to now read:

"Thus, we conclude that the exhaust of the SP2, when sampling Aquadag® with sufficient laser power, consists of small, amorphous particles formed from the homogeneous nucleation of vaporized rBC."

**P11, L23: The big caveat here is that this determination will not work if a sizeable fraction of INPs are internally mixed with BC, which is the case for burns and wildfires. Therefore, I do not agree with this statement.**

While it is true the BC from burns and wildfires may be internally mixed with non-BC components, the authors would argue that it is unknown if a sizable fraction of INPs are internally mixed with rBC as the reviewer suggests. See the General Comments section for more detail.

**P17, Fig. 3: Please indicate in the caption that particle concentrations were kept constant at 600 cm-3**

Done. Figure 3 Caption now reads:

"Size distribution of the SP2 exhaust from 125-nm Aquadag® particles at a concentration of 600 cm-3 under different sheath-to-sample ratios. For reference, the size distribution of a 125-nm Aquadag® particle at a sheath-to-sample ratio of 4.9:1 is shown in black."

**P18, Fig. 4: Please indicate in the caption what sheath:sample air flow rate ratio was used for these measurements."**

Done. See comment above.

**P21 and 22, Figs. 7 and 8: Please provide an explanation here or in the main text of exactly what the error bars represent.**

Done. The following sentence was added to Figure 7 and 8 Captions:

"The errors bars represent one standard deviation of the mean from at least three measurement periods."

**P22, Fig. 8: X-axis label missing (even if one can still kind of figure out what is shown).**

The meaning of the x-axis changes depending on the symbols in the legend. The authors, upon reviewing, agree that it is nonetheless confusing not to label the axis. Thus, we have changed the x-axis label to "Contribution or Fraction"

\* References

Michelsen, H. A., Schulz, C., Smallwood, G. J. and Will, S.: Laser-induced incandescence: Particulate diagnostics for combustion, atmospheric, and industrial applications, Progress in Energy and Combustion Science, 51, 2–48, doi:10.1016/j.pecs.2015.07.001, 2015.

**Author's Response References**

- Baumgardner, D., Popovicheva, O., Allan, J., Bernardoni, V., Cao, J., Cavalli, F., et al. (2012). Soot reference materials for instrument calibration and intercomparisons: A workshop summary with recommendations. *Atmospheric Measurement Techniques*, 5(8), 1869–1887. https://doi.org/10.5194/amt-5-1869-2012
- DeCarlo, P. F., Slowik, J. G., Worsnop, D. R., Davidovits, P., & Jimenez, J. L. (2004). Particle Morphology and Density Characterization by Combined Mobility and Aerodynamic Diameter Measurements. Part 1: Theory. *Aerosol Science and Technology*, 38(12), 1185– 1205. https://doi.org/10.1080/027868290903907
- Levin, E. J. T., McMeeking, G. R., DeMott, P. J., McCluskey, C. S., Stockwell, C. E., Yokelson, R. J., & Kreidenweis, S. M. (2014). A New Method to Determine the Number Concentrations of Refractory Black Carbon Ice Nucleating Particles. *Aerosol Science and Technology*, 48(12), 1264–1275. https://doi.org/10.1080/02786826.2014.977843
- Moteki, N., & Kondo, Y. (2010). Dependence of Laser-Induced Incandescence on Physical Properties of Black Carbon Aerosols: Measurements and Theoretical Interpretation. *Aerosol Science and Technology*, 44(8), 663–675. https://doi.org/10.1080/02786826.2010.484450

---

## Author Comment (AC2) · 5 Apr 2018

The reviewer's comments are bolded and italicized while our comments are in plain text.

Response to Reviewer #2

*This paper presents a characterisation of an SP2 instrument that is to be used as a pre-filter for ice nucleation experiments. It presents a series of careful experiments to examine the evaporation of rBC by the incandescence laser as a function of laser power and shows that even at low laser powers an ultrafine particle population is observed in the exhaust of the instrument. TEM analysis shows that this is due to nucleation and not fragmentation. The conclusion is that a laser power of 930 nW is required for complete vaporization. The effect of black carbon on the ice nucleation of a range of standard proxies was carried out and it is shown that only when black carbon is internally mixed with the proxy does its IN efficiency change.*

*These conclusions therefore allow the effectiveness of black carbon particles as IN to be separated from other IN as long as the population is externally mixed. Where significant internal mixing has taken place then its use as a probe may be less unequivocal. This characterisation is important and is certainly worthy of publication in AMT. The paper concludes that the results bolster confidence in the method to separate the effects of rBC in IN experiments and cites wildfires as an important area.*

*I would question whether the tone should be so optimistic since significant dust is often present in wildfires and the mixing of rBC and dust in the near field of fires is not well characterised. My interpretation of the results presented here is that this is a potential shortcoming that cannot easily be overlooked nor tackled when using ambient data. This should be included in the final discussion and ant methods to identify its influence identified.*

*The paper is largely very well written in my view and the figures are clear and understandable. There are one or two places where the text could be made more readable and these are identified below.*

The authors would first like to thank Reviewer #2 for his/her insightful comments, which have improved the clarity and utility of this manuscript.

The authors agree with the reviewer about the tone of the discussion. In light of this comment and a general comment by Reviewer #1, we have amended the final paragraph of Section 3.4 to now read:

"When the INP proxies are internally mixed with rBC, we do see a reduction in INP concentrations due to the SP2 laser. Thus, INPs internally mixed with rBC generally cause overestimations of the rBC contribution to INP concentrations in the SP2-CFDC. To account for this, we determined the fraction of "attached-type" particles (section 2.3). Interestingly, we find that the attached-type fraction correlates well with the number fraction of deactivated INP for the internally mixed NX-illite case. The INEs in NX-illite are refractory, and therefore will not be fully vaporized when an attached rBC particle is heated to 4000 K; thus, scattering material

would traverse SP2 laser and these particles would appear as "attached" in the SP2 analysis. From Figure 8, we see that 97% of the attached-type fraction was deactivated after exposure to the SP2 laser. This is consistent with results from Levin et al. [2014], who used a different analysis to estimate the number of attached-type particles, and found that, at minimum, 74% of mixed Aquadag®-ATD particles were deactivated as INP following exposure to the SP2 laser. Thus, for refractory INE attached to rBC, the attached-type fraction can be used to estimate the number of deactivated, rBC-containing INP.

In contrast, the attached-type fraction does not correlate with the number of deactivated INP for the internally mixed SRFA and Snomax® cases. Here, unlike NX-illite, the INEs from SRFA and Snomax® are non-refractory/heat-labile. Thus, when rBC heats to 4000 K in the SP2 laser, any attached, non-refractory INEs are completely evaporated or destroyed. To confirm this, we estimated the fraction of rBC-SRFA/Snomax® particles that were affected by the SP2-laser. During one-pot nebulization, or what we are calling the "internally mixed" scenario, some fraction of particles will be pure rBC, some fraction will be pure SRFA/Snomax®, and a final fraction will be truly internally mixed. As a conservative estimate, we assume that all particles that contain incandescing material also contain INPs. From the SP2 raw data, 96 and 76% of all Aquadag®-SRFA and Aquadag®-Snomax® particles in the "internally mixed" scenario contained incandescent material, respectively. Thus, only 96 and 76% of the particles could contain rBC that is both greater than 90 nm and physically attached to an INE. From Figure 8, the fraction of deactivated INP was 90% for "internally mixed" Aquadag®-SRFA and 69% for "internally mixed" Aquadag®-Snomax® particles. Thus, 94% of the incandescent Aquadag®-SRFA particles and 91% of the incandescent Aquadag®-Snomax® particles were deactivated following exposure to the SP2 laser. From the combined above analyses, we believe that heating rBC to its vaporization temperature of ~4000 K will destroy >90% of any physically attached INE. Therefore, while this technique cannot tell us the number of rBC INP, it can tell us the number of rBC-containing INP. Thus, we recommend using the terminology "rBC-containing contribution," instead of the previously used "rBC contribution."

***It might be worth commenting in the discussion on whether the nucleation arises from the organic matter evaporating and then re-nucleating as well as the core. Whilst this is not important in your experiment since the particles were nearly all composed of rBC but for particles with significant coating, such as biomass burning particles it could a much bigger effect. One might also expect significant condensational growth of the nucleated particles in these conditions. Given the conclusions are focused on the use of the instrument for investigating the IN effectiveness of biomass burning this is worth including.***

The authors agree that organic matter coating an rBC core would evaporate, and could re-nucleate or condense on nucleated particles. This, in theory, could affect the ice nucleation effectiveness of externally mixed INP. In general, however, it is not thought that non-biologically sourced, liquid organics would affect the ice nucleation of insoluble particles. Interestingly, a coating of amorphous elemental carbon may affect externally mixed INP, but we saw no evidence of this from the externally mixed experiments. Nonetheless, we have added the following statement to Section 3.4:

"Finally, condensation of evaporated organics or vaporized rBC may coat and deactivate externally mixed INPs; while this is a lesser concern for non-biologically sourced liquid organics (Prenni et al., 2009; Schill et al., 2016), a coating of solid, amorphous carbon could cause a significant reduction in ice nucleation ability."

***Page 3 line 36: This line seems to alternate between singular and plural (finesses/face)***

Agree. "Finesses" has now been changed to "finesse."

***Page 4 lines 5-6: s in figure and S in text***

The authors thank the reviewer for carefully reading the manuscript for typos. In this case, however, the "s" in the equation is actually a capital "S."

***Page 4 line 25: The description of the positions in not clear. Is this the time delay from the first point the signal passes the reference threshold? If so, which detector? This needs to be clear.***

The authors agree that the description of the positions could be clarified. The sentence in question has been changed to the following:

"To approximate the number of attached-type particles, we take the difference between the scattering peak's half-decay position and incandescent peak's half-decay positions"

***Page 4: line 28: depend(a)nt dependent***

Thank you. Corrected.

***Page 6 line 7: A lesson learned from bitter experience I suspect***

This was indeed a lesson learned from experience, and hopefully one that is useful for others replicating this technique.

***Page 7 line 4: "of at least At"***

Thank you. "At" was changed to "at."

***Figure 3: caption state the laser power used***

The authors agree that this adds clarity to Figure 3. We added the following sentence to the figure caption:

"The absolute SP2 laser power was 1290 nW/(220-nm PSL)."

***Page 9 lines 24-27: I do not feel that "rBC containing contribution" in figure 8 is well described at all. What is "the effect" on laser power? and what does the scale on the x axis of the figure represent? This needs clarification.***

Upon review, the authors agree that the results in Figure 8 is difficult to comprehend because there are several key pieces of information. As suggested by the reviewer, we have removed the pure proxies from Figure 8, and added the laser's effect on the ice nucleation properties of each pure proxy into Figure 7. We now feel both that Figures 7 and 8 effectively communicate their main results.

Additionally, the meaning of the x-axis changes depending on the symbols in the legend. The authors, upon reviewing, agree that it is nonetheless confusing not to label the axis. Thus, we have changed the x-axis label to "Contribution or Fraction"

***Page 10 line 16 of (the) SP2 laser***

Corrected. Thank you.

***Page 10 line 26: particles***

Corrected. Thank you.

Author's Response References

Prenni, A. J., Petters, M. D., Faulhaber, A., Carrico, C. M., Ziemann, P. J., Kreidenweis, S. M., & DeMott, P. J. (2009). Heterogeneous ice nucleation measurements of secondary organic aerosol generated from ozonolysis of alkenes. *Geophysical Research Letters*, *36*(6), 1–5. https://doi.org/10.1029/2008GL036957

Schill, G. P., Jathar, S. H., Kodros, J. K., Levin, E. J. T., Galang, A. M., Friedman, B., et al. (2016). Ice nucleating particle emissions from photochemically-aged diesel and biodiesel exhaust. *Geophysical Research Letters*, n/a-n/a. https://doi.org/10.1002/2016GL069529